# The Analysis, Description, and Examination of the Maize LAC Gene Family’s Reaction to Abiotic and Biotic Stress

**DOI:** 10.3390/genes15060749

**Published:** 2024-06-06

**Authors:** Tonghan Wang, Yang Liu, Kunliang Zou, Minhui Guan, Yutong Wu, Ying Hu, Haibing Yu, Junli Du, Degong Wu

**Affiliations:** 1College of Agriculture, Anhui Science and Technology University, Fengyang 233100, China; 18767401521@163.com (T.W.); zkl151231@163.com (K.Z.); wuyutong1385@163.com (Y.W.); huying2191@163.com (Y.H.); yuhb@ahstu.edu.cn (H.Y.); adu83419@163.com (J.D.); 2College of Resource and Environment, Anhui Science and Technology University, Fengyang 233100, China; 18298115131@163.com (Y.L.); mhbelucky@163.com (M.G.)

**Keywords:** maize, LAC, gene family, abiotic and biotic stresses, expression pattern analysis

## Abstract

Laccase (LAC) is a diverse group of genes found throughout the plant genome essential for plant growth and the response to stress by converting monolignin into intricate lignin formations. However, a comprehensive investigation of maize laccase has not yet been documented. A bioinformatics approach was utilized in this research to conduct a thorough examination of maize (*Zea mays* L.), resulting in the identification and categorization of 22 laccase genes (ZmLAC) into six subfamilies. The gene structure and motifs of each subgroup were largely consistent. The distribution of the 22 LAC genes was uneven among the maize chromosomes, with the exception of chromosome 9. The differentiation of the genes was based on fragment replication, and the differentiation time was about 33.37 million years ago. ZmLAC proteins are primarily acidic proteins. There are 18 cis-acting elements in the promoter sequences of the maize LAC gene family associated with growth and development, stress, hormones, light response, and stress response. The analysis of tissue-specific expression revealed a high expression of the maize LAC gene family prior to the V9 stage, with minimal expression at post-V9. Upon reviewing the RNA-seq information from the publicly available transcriptome, it was discovered that *ZmLAC5*, *ZmLAC10*, and *ZmLAC17* exhibited significant expression levels when exposed to various biotic and abiotic stress factors, suggesting their crucial involvement in stress responses and potential value for further research. This study offers an understanding of the functions of the LAC genes in maize’s response to biotic and abiotic stress, along with a theoretical basis for comprehending the molecular processes at play.

## 1. Introduction

Yoshida found laccases (LACs; EC 1.10.3.2) in a lacquer tree in Japan in 1883 [1] and classified them as part of the copper blue protein oxidase family [2]. LACs have been isolated and purified from various plant sources, such as maple and pine trees [3,4,5]. These enzymes are typically comprised of three conserved copper oxidase structural domains associated with four copper ions: a mononuclear blue copper ion (Cu1) situated at the T1 (copper ion (Cu1)) site, exhibiting a characteristic blue hue, and a trinuclear copper cluster at the T2/T3 sites, consisting of one T2 (copper ion (Cu2)) and two T3 (copper ions (Cu3)) [6,7,8]. LACs have the ability to oxidize a variety of fragrant and non-fragrant compounds when oxygen is present, such as dianilines, bisphenols, hydroxylamines, and monolignols [9]. They are prevalent across various organisms, such as bacteria [10], fungi [8,11], plants [12], and insects [13], with plant LAC forming a distinct phylogenetic branch [14]. Notably, LACs have been characterized in a broad spectrum of plant species [12], including the lacquer tree (*Toxicodendron vernicifluum*) [15], *Arabidopsis* (*Arabidopsis thaliana*) [16,17], Purple Falsebrome (*Brachypodium distachyon*) [18], oilseed rape (*Brassica napus*) [19], cotton (*Gossypium hirsutum*) [20], poplar (*Populus duccarpa*) [21], rice (*Oryza sativa*) [22,23], sorghum(*Sorghum bicolor*) [24], perennial ryegrass (*Lolium perenne*) [25], sugarcane (*Saccharum officinarum*) [26], tobacco (*Nicotiana tabacum*) [27], and yellow poplar (*Liriodendron tulipifera*) [28].

Many studies have investigated fungal LAC, focusing on diverse areas such as pigment synthesis, growth and development, host infection, the defense response, and xenobiotic degradation [29,30,31]; however, recent studies suggest that our knowledge, regarding the LACs in higher plants is still limited. LACs are essential for controlling a wide range of biological functions, such as rice seed development, stone cell creation, secondary cell wall production, lignification, pigmentation, flavonoid formation, cotton fiber excellence, cell elongation, and fruit preservation after harvesting [17,32,33,34,35,36,37]. LAC is involved in how plants react to both living and non-living pressures [38,39]. For example, cotton *GhLAC15* is associated with a resistance to Verticillium wilt, while *GhLAC4* enhances G-lignin synthesis and thickens the cell wall to combat Fusarium graminearum infection [40,41]. Similarly, maize *ZmLAC1* and Lacertae *RvLAC2* are involved in responding to salt and repairing wounds, respectively [42,43], while the microRNA528-ZmLac3 system controls maize’s ability to tolerate low phosphate levels [44].

Maize (*Zea mays* L.), commonly referred as corn, is a staple crop globally, with significant importance in the food and health industries [45,46]. Inputs are an important way to improve maize yield and responsiveness. Despite the clear advantages of newly developed varieties, the productive ability of maize genotypes has still been limited. A variety of factors, including climate, pests, soil properties, solar radiation, field management practices, seed quality, and genetic potential, can significantly impact a maize genotype’s productivity and yield potential [47]. Thus, it is essential to identify the genes in maize that are resistant to stress and to create varieties that can withstand stress in order to enhance both quality and yield. An essential approach to understanding gene origin and predicting gene function is the analysis of gene families [48,49,50]. Various gene families, such as C2H2 [51], GS [52], LACS [53], FAR1/FHY3 [54], and SWEET [55], have been characterized in maize using high-quality genomic data. Nevertheless, previous investigations of maize LAC family genes were solely based on genomic information from the maize B73_V4 variety [43,56], resulting in an incomplete genome identification of the maize LAC family genes. The abiotic and biotic stress expression patterns of maize LAC family genes have not previously been examined in research, limiting our understanding of their biological functions.

Bioinformatics methods were utilized in this research to identify LAC family genes within the maize B73_V5 genome, aiming to address knowledge gaps regarding maize LAC genes. The examination of maize LAC family members included an analysis of their physical and chemical characteristics, chromosome position, genetic makeup, enriched functions, evolutionary relationships, and collinear relationships. The analysis of transcriptome sequencing data was accomplished using large-scale data from maize transcriptome sequencing based on genomic information from the maize B73_V5 variety. It was possible to gain an understanding of the biological function of the maize LAC gene family by studying the expression and patterns of LAC genes in different tissues under stress. These findings provide important information on how the LAC family members in maize are regulated during abiotic stress and also highlight genes that could be used to improve stress tolerance in maize through molecular breeding techniques.

## 2. Materials and Methods

### 2.1. Identification and Chromosome Mapping of the LAC Family Genes in Maize

During a literature search, the protein sequences of *Arabidopsis* laccase (AtLAC1-AtLAC17) were acquired from the *Arabidopsis* Information Repository (https://www.Arabidopsis.org/; accessed on 22 January 2024) to identify members of the maize LAC gene family [57]. The HMM model files of the LAC gene family was downloaded (PF00394, PF0773, and PF07732) from the Pfam database (http://pfam.xfam.org/; accessed on 22 January 2024) [58]. The Maize B73_V5 genome sequence and GFF3 file were acquired from NCBI (https://www.ncbi.nlm.nih.gov/datasets/genome/; accessed on 22 January 2024), while a protein database was established on a local server [59].

The sequence containing the LAC protein domain was searched for on the HMMER website (https://www.ebi.ac.uk/Tools/hmmer/; accessed on 22 January 2024) [60] using the local BLASTP (version 2.13.0) program. A threshold of E < 1 × 10^−20^ was utilized to filter potential gene candidates, resulting in the acquisition of the sequence data of the candidate protein. The online tools InterPro (https://www.ebi.ac.uk/interpro/; accessed on 22 January 2024) [61], SMART (http://smart.embl-heidelberg.de; accessed on 22 January 2024) [62], and CDD (https://www.ncbi.nlm.nih.gov/Structure/bwrpsb/bwrpsb.cgi; accessed on 22 January 2024) were used to verify the potential LAC gene sequence [63].

By removing the comparable SKU5, the Cupredoxin superfamily, and L-ascorbate oxidase, it was anticipated that the enzymes containing standard copper oxidase domains were potential laccase members that could be used to identify the individuals of the LAC gene family in maize. The maize LAC gene family’s chromosomal distribution was mapped using TBtools software (version 2.067) [64]. The sequence is situated on the chromosome corresponding to ‘Chr N’, where N represents a number between 1 and 10.

### 2.2. Analysis of the Physicochemical Characteristics and Conserved Motif of the LAC Gene Family in Maize

The online tool ExPASy was utilized to examine various characteristics of the maize LAC gene family, such as their amino acid count, molecular weight, isoelectric point, instability coefficient, fat index, and average hydrophilicity (https://web.expasy.org/protparam/; accessed on 23 January 2024) [65]. Maize subcellular localization was predicted for LAC in the online platform WoLF PSORT (https://wolfpsort.hgc.jp/; accessed on 23 January 2024) [66]. The maize LAC genes’ laccase superfamily domain was acquired from the NCBI database (https://www.ncbi.nlm.nih.gov/cdd; accessed on 23 January 2024) [63].

The maize LAC gene motif was anticipated and examined through the utilization of MEME software (http://meme-suite.org/; accessed on 23 January 2024) [67]. There were 10 motifs in total, with the remaining parameters set to default. The LAC gene in maize was analyzed using TBtools software to generate a diagram showing its conserved motif, domain, and exon–intron organization. Information on the structural domains of the copper oxidase (cu-oxidase-1/2/3) detected in the candidate genes was annotated using Jalview (version 2.11.3.3) software.

### 2.3. Phylogenetic Analysis of the LAC Family Genes in Maize

We elucidated the evolutionary relationship between maize LAC and other plant LACs; a dataset comprising 17 LAC proteins from *Arabidopsis* [17], 30 from rice [23], 27 from sorghum [24], and 49 from soybean was assembled from studies on these respective LAC gene families [12]. The LAC protein sequences from maize, *Arabidopsis*, rice, sorghum, and soybean were aligned using the default parameters of MEGA software (version 10). A tree was generated to assess the robustness of the findings by employing the neighbor-joining technique and conducting 1000 bootstrap resampling iterations. Evolview software was employed for the visualization and enhancement of the phylogenetic trees (https://www.evolgenius.info/evolview; accessed on 30 January 2024) [68].

### 2.4. Collinearity Analysis of the LAC Family Genes in Maize

The creation of new gene families and functions is primarily driven by gene duplication and diversification. We utilized TBtools software to identify orthologous maize genes across other species. This study analyzed the alignment of maize LAC genes within the same species and compared them to the LAC genes in dicotyledonous plants (*Arabidopsis thaliana* and soybean) and monocotyledonous plants (rice and sorghum). Their gene duplication patterns were investigated using the MCScanX plugin in TBtools [69], followed by calculating the synonymous substitution rate (Ks) and the nonsynonymous substitution rate (Ka). A Ka/Ks ratio of less than 1 indicates purifying selection, a ratio of 1 represents neutral selection, and a ratio greater than 1 suggests positive selection [59,60,61,62,63,64,65,66,67,68,69,70,71,72]. The approximate time of divergence was determined using the equation T = Ks/(2λ), where λ is set at 6.5 × 10^−9^ [73,74].

### 2.5. Functional Analysis of the LAC Family Genes in Maize

In order to clarify the operational characteristics of the LAC genes, we employed the web-based tool ShinyGO v0.741 to conduct a Gene Ontology (GO) Enrichment Analysis (http://bioinformatics.sdstate.edu/go74/; accessed on 1 February 2024) [75]. By inputting putative candidate proteins or gene IDs, GO annotations were obtained for maize. A *p*-value (FDR) < 0.01 was used as the significance threshold for assessing the GO enrichment level, with visualization carried out through the online tool WeChat (https://www.bioinformatics.com.cn/; accessed on 1 February 2024) [76]. Furthermore, PlantCare (https://bioinformatics.psb.ugent.be/webtools/plantcare/html/; accessed on 1 February 2024) was utilized to predict the cis-regulatory elements located in the promoter region in the 2000 base pairs before the maize LAC gene family [77].

### 2.6. Re-Evaluation of the Large-Scale Data from RNA-Seq for Maize Transcriptome Sequencing

The corn transcriptome sequencing information from the Sequence Read Archive (SRA) was transformed into a fastq file using the SRA toolkit (version 3.7.1). FastQC software was used to evaluate the quality of the Fastq data (https://github.com/s-andrews/FastQC; accessed on 5 March 2024) [78]. The Fastq data were filtered, and contaminated and low-quality sequences were eliminated using Trimmomatic software (version 0.40) to create high-quality data. The processed information was compared to maize B73_V5 genetic material using the Kallisto Super Wrapper add-on in TBtools to produce Transcripts per Million documents, allowing for a calculation of its gene expression quantities. Finally, the gene expression patterns were visualized utilizing the HeatMap plugin in TBtools, which depicts the expression profiles of individual genes.

### 2.7. Analysis Was Conducted on the Expression Patterns of the LAC Gene Family in Maize across Various Tissues

Data from different varieties of corn (SRP010680) [79] were obtained from the National Center for Biotechnology Information (NCBI) website (https://www.ncbi.nlm.nih.gov/sra; accessed on 15 February 2024). Subsequently, the transcriptome data were reviewed by aligning them with the maize B73_V5 genome. The maize LAC gene family expression patterns across various tissues were illustrated using an expression heat map created with TBtools software.

### 2.8. Analysis of the Expression of LAC Family Genes in Maize under Both Abiotic and Biotic Stress Conditions 

Data on maize gene expression were obtained from the NCBI repository, covering a range of environmental challenges such as salinity, waterlogging, extreme temperatures, and drought, in addition to diseases like leaf spot and smut and pests like aphids and sugar beet noctuid moths. The B73_V5 maize genome data were utilized to reanalyze this information, resulting in the creation of a heat map for the LAC gene family with the TBtools program.

## 3. Results

### 3.1. An Overview of the Maize LAC Gene Family Members

Using the maize B73_V5 genomic data, the *Arabidopsis laccase* (AtLAC1–AtLAC17) sequences were subjected to a BLAST analysis, and the LAC gene family’s HMM model was obtained from the Pfam database. This analysis yielded thirty-nine putative genes, with two classified as cupredoxin superfamily proteins, four as L-ascorbate oxidases, seven as SKU5 homologs, four as uncharacterized proteins, while the remaining twenty-two contained characteristic copper oxidase domains indicative of potential laccases (see Appendix A for details). Nevertheless, equivalent L1–L4 signature sequences are present in members of the maize laccase family, and these sequences distinguish laccases from the broader group of multicopper oxidases (see Appendix A for details). Amino acids that may be associated with copper binding, such as 10 histidines and 1 cysteine, along with axial ligands for methionine or leucine, can be found in four conserved areas (refer to Appendix A for more information).

Proteins serve as a precise reflection of phenotypic regulation and represent the fundamental manifestation of biological processes. An examination of their physical and chemical characteristics showed that the maize LAC genes had coding sequences (CDSs) varying in size from 1473 to 2001 base pairs, which translated to 490–666 amino acids, and had molecular weights ranging from 53.78–71.36 kilodaltons. The amino acid count demonstrated a direct correlation with molecular weight. The theoretical isoelectric point ranged from 5.41 to 9.86, indicating predominantly acidic characteristics. The instability coefficient varied from 29.13 to 55.16, with 7 LAC proteins classified as stable (instability index > 40) and 15 as unstable (instability index < 40). The range of the aliphatic index was between 71.89 and 90.00. Notably, the average hydrophobicity of 14 LAC proteins was negative, suggesting hydrophilicity, while 8 LAC proteins exhibited positive average hydrophobicity, indicative of hydrophobicity (see Appendix A for details). The subcellular localization predictions indicated that the maize LAC proteins primarily function in chloroplasts as opposed to vesicles (Figure 1).

### 3.2. Mapping the LAC Family Genes in Maize Using Chromosome Analysis

The chromosomal localization analysis of the 22 genes within the maize LAC family facilitated the construction of a distribution map depicting the arrangement of the maize LAC gene family across chromosomes (Figure 2). Figure 2 illustrates this distribution, revealing an uneven dispersion of the 22 LAC genes across all maize chromosomes, except chromosome 9. Chromosome 3 had the most LAC genes, a total of seven, whereas chromosome 4 had six LAC genes, the second highest number. In contrast, chromosomes 2, 5, 6, 7, and 10 each contained a solitary LAC gene. Notably, the LAC genes residing on chromosomes 3 and 4 predominantly exhibited clustering tendencies.

According to the evolutionary relationships between the family members and chromosome locations, 11 gene pairs were involved in gene replication events, of which two were tandem duplications, accounting for 18.18%, and nine were fragment duplications, accounting for 81.82%. Thus, the expansion of the maize LAC family occurred through tandem duplication and fragment duplication, and fragment duplication occurred more frequently, indicating that gene duplication between different chromosomes is the main method used to expand the maize LAC gene family. *ZmLAC7* and *ZmLAC20* had three fragment replications, indicating that *ZmLAC7* and *ZmLAC20* were the most active and contributed the most to gene expansion. The gene differentiation formed by fragment replication occurred over 11.34–46.06 million years, and tandem replication occurred over 23.63–56.48 million years. The Ka values for 11 duplicated gene pairs were lower than the Ks values, suggesting synonymous substitution provides an advantage. The Ka/Ks ratio for the 11 pairs of genes has been adjusted to 1, indicating that these pairs have undergone considerable purifying selection and have not yet diverged in their functions (see Appendix A for details).

### 3.3. Phylogenetic Tree Analysis of the LAC Gene Family in Maize

In order to completely explain the similarities and functions of the maize LAC gene family, the maize LAC genes that were found were grouped with LAC gene family members from *Arabidopsis*, soybean, rice, and sorghum through multiple sequence comparison to create a phylogenetic tree (Figure 3). Based on the *Arabidopsis* LAC gene family classification results, the maize LAC gene family has been divided into six subgroups. Among the subgroups, V exhibited the greatest quantity of maize LAC genes, totaling 10. Six maize LAC genes were identified in Subgroup I, four in Subgroup III, and two maize LAC genes were identified in Subgroup IV. No maize LAC genes were observed in Subgroups II or VI, and only one *Arabidopsis* laccase (*AtLAC1*) was found in subgroup VI. Twenty percent of the plants’ total genes had orthologous relationships with thirty LAC genes. Among them, the maize–sorghum genome contained eleven pairs of orthologous genes: one pair of orthologous genes was found in maize–rice, one pair of orthologous genes was found in rice–sorghum, and one pair of orthologous genes was found in *A. thaliana*–soybean, indicating that there were more orthologous genes between maize and sorghum in their gramineous monocotyledons, and their genetic relationship was closer. There were thirty-five pairs of paralogous genes, including nineteen pairs from soybean, five pairs from rice, four pairs from maize, four pairs from *A. thaliana*, and three pairs from sorghum (see Appendix A for details).

### 3.4. An Examination of the LAC Gene Family in Maize through Comparing Gene Structure and Conserved Sequences

The maize LAC gene family’s structure was clustered and analyzed using TBtools software. Figure 4 shows that the maize LAC gene family is divided into four subfamilies: groups I, III, IV, and V (Figure 4). These findings are in line with the LAC genes in *Arabidopsis*, sorghum, soybean, and rice, with groups II and VI lacking any members from the maize LAC gene family. According to the maize LAC gene family structure map, all gene members contained introns, exons, and UTR structures. They had 2–5 introns and 3–6 exons. Similar to the laccases in other plants, it was inferred that these laccases may be more conserved in different plants.

Using the MEME website, with 10 motifs as the parameters for prediction, the results revealed that *ZmLAC6* and *ZmLAC14* lacked two motifs each, while the remaining 20 members of the gene family all contained 10 motifs and exhibited the same arrangement order. This observation suggests that the gene family members possess good integrity and are relatively conserved. All 22 maize LAC genes contained the three typical structural domains present in laccase: Cu-oxidase-3, Cu-oxidase-1, and Cu-oxidase-2. TBtools and the MEME web application were utilized for the examination of conserved patterns in maize LAC proteins (see Appendix A for details). Maize LAC possessed conserved copper-binding domains, with different subgroups of LAC proteins exhibiting distinctly conserved sequences; the conserved sequences within the same subgroup were identical.

### 3.5. An Examination of the Functional Aspects of the LAC Gene Family in Maize

The promoter sequence of maize LAC genes contained 18 distinct cis-acting elements that were identified (Figure 5). A significant proportion (33.1%) of them are related to the plant’s light response, followed by those related to methyl jasmonate responsiveness (20%). Furthermore, specific cis-regulatory components have been identified to be involved in the reaction to various hormones (like gibberellin, abscisic acid, auxin, and salicylic acid), stress factors (such as anaerobic conditions, low temperature, and drought), defense responses, the control of seed-specific genes, the growth of meristems, and the expression of endosperm genes. Various cis-acting elements were found in various promoter regions of LAC genes, indicating the diverse functions of LAC genes in maize’s growth and development. The maize LAC genes were functionally annotated using Weishengxin for visualization and analysis, with data from the ShinyGO v0.741: GO Enrichment Analysis. Of the 22 maize LAC genes examined, 11 were successfully annotated (Figure 6). The molecular function enrichment analysis revealed prominent associations with phenylpropanoid decomposition and lignin catabolism (Figure 6A). Additionally, their biological function enrichment highlighted a notable enrichment in oxidoreductase activity and copper ion binding (Figure 6B).

### 3.6. Collinearity Analysis of the LAC Family Genes in Maize

A collinearity analysis was conducted between maize LAC genes and those of dicotyledonous plants (*A. thaliana* and soybean) (Figure 7A) and monocotyledonous plants (rice and sorghum) (Figure 7B) using TBtools software. A total of 96 interspecific collinear relationships were identified. Specifically, the researchers discovered 8 collinear links between maize and *A. thaliana*, 28 between maize and rice, 23 between maize and sorghum, and 37 between maize and soybean. Notably, *ZmLAC2* and *ZmLAC10* exhibited the highest collinearity, with 14 relationships, followed by *ZmLAC21* with 12. Additionally, *ZmLAC3*, *ZmLAC5*, *ZmLAC8*, *ZmLAC9*, and *ZmLAC12* were collinear, while *ZmLAC14* and other species were not. These analyses were conducted to determine the Ka, Ks, and Ka/Ks ratios for each pair of genes. The findings indicated that the average Ka, Ks, and Ka/Ks ratios of the maize LAC genes’ consecutive repetitions were higher compared to their fragment repetitions. The Ks value of the maize gene pairs with *Arabidopsis* and rice homologs was notably greater than 2, while the maize gene pairs exhibited Ka, Ks, and Ka/Ks values below 1 in comparison to their *Arabidopsis*, soybean, rice, and sorghum homologs (Figure 7C–E). Approximately 39.83 million years ago, the maize LAC gene experienced duplication events for both tandem and fragment repeat genes, while the latter occurred around 33.37 Mya. Maize diverged from *Arabidopsis*, soybean, rice, and sorghum around 212.55, 228.93, 39.32, and 30.37 million years ago, respectively.

### 3.7. Analysis Was Conducted on the Expression Patterns of the LAC Family Genes in Maize in Various Tissues

Using public transcriptome sequencing data from diverse maize tissues (SRP01680) [79], a new transcriptome sequencing analysis was conducted with maize B73_V5 genome information, resulting in an expression heat map demonstrating the LAC gene family across a variety of tissues (Figure 8). The maize LAC gene family exhibited high expression levels up to the V9 Thirteenth Leaf stage, followed by a notable decrease in expression thereafter. Notably, *ZmLAC8*, *ZmLAC9*, *ZmLAC10*, *ZmLAC18*, *ZmLAC19*, and *ZmLAC20* displayed peak expression levels at the V9 Eleven Leaf, V9 Thirteenth Leaf, and V9 Immature Leaves stages, coinciding with the jointing stage of maize growth, where extensive laccase activity is required for xylem synthesis. In contrast, *ZmLAC1, ZmLAC3, ZmLAC4, ZmLAC11, ZmLAC12, ZmLAC15, ZmLAC16, ZmLAC17,* and *ZmLAC22* were minimally expressed across most maize tissues. Notably, *ZmLAC5* exhibited a normal distribution pattern before 12DAP_Whole seed, with subcellular localization in the plasmodesmata and phylogenetic classification in subgroup V, suggesting a specialized function in maize resistance mechanisms.

### 3.8. Analysis of Expression Patterns of Maize LAC Family Genes under Abiotic Stress 

The examination of the transcriptome sequencing data of maize subjected to different environmental stresses, including salt stress (PRJNA414300) [80], waterlogging stress (PRJNA606824) [81], high- and low-temperature stress (PRJNA645274) [82], and drought stress (PRJNA576545) [83], was carried out using maize B73_V5 genomic information. From different abiotic stress responses, a heat map of the maize LAC gene family was generated (Figure 9). Overall, the expression of the maize LAC gene family was high under waterlogging stress but lower under drought stress. *ZmLAC5*, *ZmLAC10*, and *ZmLAC17* showed elevated levels of expression when exposed to different abiotic stress factors (Figure 9A). The expression of three LAC genes (*ZmLAC10*, *ZmLAC17*, and *ZmLAC22*) was notably elevated under salt stress in both salt-sensitive and salt-tolerant maize varieties in comparison to their controls. Three LAC genes (*ZmLAC5*, *ZmLAC15*, and *ZmLAC17*) were significantly upregulated under waterlogging stress, while seven LAC genes (*ZmLAC2*, *ZmLAC3*, *ZmLAC7*, *ZmLAC9*, *ZmLAC10*, *ZmLAC18*, and *ZmLAC19*) were significantly downregulated (Figure 9B). Four LAC genes (*ZmLAC5*, *ZmLAC9*, *ZmLAC17*, and *ZmLAC22*) showed comparable expression profiles in response to both high- and low-temperature stress, with *ZmLAC5* being notably upregulated, while *ZmLAC9*, *ZmLAC17*, and *ZmLAC22* were significantly downregulated (Figure 9C). Under drought stress, only *ZmLAC4* was significantly upregulated (Figure 9D), while other genes remained unchanged.

### 3.9. Analysis on How Maize LAC Family Genes Are Expressed during Biotic Stress

Transcriptome sequencing analysis was carried out on data from maize leaf spot (*Mycosphaerella maydis*) (PRJNA436207) [84], smut (*Sphacelotheca reiliana*) (PRJNA673988) [85], aphid (*Rhopalosiphum maidis*) (PRJNA295410) [86], and beet armyworm stress (*Spodoptera exigua*) (PRJNA625224) [87], in relation to the maize B73_V5 genome information. A heat map showing the LAC gene family’s expression responses to biotic stressors in maize was created (Figure 10). Notably, the maize LAC gene family exhibited the most pronounced expression levels under smut stress, with comparatively lower expression levels observed under the aphid treatment. *ZmLAC5* and *ZmLAC10* displayed relatively high expression levels across various stressors.

The *ZmLAC8* gene was upregulated in plants resistant and susceptible to maize gray spot disease, with a more pronounced upregulation observed in resistant plants (Figure 10A). In contrast, *ZmLAC5* was downregulated in resistant and susceptible plants, with a more significant downregulation observed in susceptible plants. Eight maize LAC genes (*ZmLAC5*, *ZmLAC7*, *ZmLAC8*, *ZmLAC9*, *ZmLAC10*, *ZmLAC15*, *ZmLAC19*, and *ZmLAC20*) exhibited significant upregulation under smut stress (Figure 10B), with *ZmLAC7* and *ZmLAC9* upregulated in six maize strains. No significant differences in expression were observed under aphid stress (Figure 10C), while four LAC genes (*ZmLAC10*, *ZmLAC15, ZmLAC17*, and *ZmLAC22*) were significantly upregulated under beet moth stress (Figure 10D).

### 3.10. Analysis of the Regulatory Mode of the Maize LAC Family Genes under Abiotic and Biotic Stressors

An analysis of maize LAC gene expression profiles under various biotic and abiotic stressors was carried out and heat maps were created to visualize the differentially expressed genes (Figure 11). Overall, maize LAC genes played a role in reacting to both abiotic and biotic stress in maize, displaying different levels of activity in response to abiotic stress and heightened activity in response to biotic stress. The maize LAC genes exhibited peak expression levels in response to waterlogging and black tassel stress, with the highest number of differentially expressed genes observed under waterlogging conditions. However, the expression of the maize LAC genes was at its least significant under the drought and aphid stressors.

The differential expression analysis of several maize LAC genes revealed that *ZmLAC2, ZmLAC3*, *ZmLAC4*, and *ZmLAC18* were exclusively differentially expressed under abiotic stress. Conversely, *ZmLAC8*, *ZmLAC6*, and *ZmLAC20*, were exclusively differentially expressed under biotic stress. Additionally, *ZmLAC5*, *ZmLAC7*, *ZmLAC9*, *ZmLAC10*, *ZmLAC17*, *ZmLAC19*, and *ZmLAC22* were differentially expressed under both stress types, but with distinct expression patterns. *ZmLAC15* was significantly upregulated under both abiotic and biotic stress conditions, suggesting its active participation in the stress response. Therefore, *ZmLAC15* is a potential key candidate gene for further investigation. There was no difference in the expression of the remaining seven LAC genes under any of the stressors. These findings will provide a basis for upcoming studies on the molecular biology of the maize LAC gene family in reaction to environmental and living factors.

## 4. Discussion

The progress and extensive use of genome sequencing technologies have resulted in the release of a growing volume of plant genomic sequencing data [88]. Consequently, the identification of key gene families has been a focus of recent research efforts. During the identification of gene families, sequence homology-based methods may miss novel LAC genes with low sequence similarity or include non-LAC genes with high similarity in certain domains. The LAC gene family plays a pivotal role in plant growth regulation, the defense response, and stress tolerance [39]. Extensive investigations of the LAC genes have been conducted in various plant species, such as the lacquer tree [15], *Arabidopsis* [16,17], Purple Falsebrome [18], oilseed rape [19], cotton [20], poplar [21], rice [22,23], sorghum [24], perennial ryegrass [25], sugarcane [26], tobacco [27], and yellow poplar [28].

Based on the genomic data of the maize B73_V4 genome, several studies have identified maize LAC genes, but comprehensive analyses of their LAC physicochemical properties and evolutionary relationships are lacking. Using bioinformatics methods, this research investigated the fundamental characteristics, evolutionary relationships, and alignment of the maize laccase gene family to understand their functional roles, similar to studies on *Arabidopsis laccases* [17]. The functional roles of the maize LAC gene family are not well understood because their expression profiles under different abiotic and biotic stress conditions have not been thoroughly explored. The maize LAC gene family was identified through the use of maize B73_V5 genome data in this study. Examining transcriptome sequencing data from the maize B73_V5 genome revealed the LAC gene family’s expression patterns in different tissues and stress environments. These findings will provide valuable resources for further studies on the functional roles of maize LAC genes and lay a theoretical foundation for improving maize stress tolerance through molecular breeding strategies.

In this research, it was discovered that maize has 22 LAC gene members, a number that sets it apart from other plant species like *Arabidopsis*, rice, soybean, and sorghum, which have 17, 30, 49, and 27 LAC genes, respectively. The variability in the number of LAC genes in different plant species suggests the presence of species-specific differences. The 22 maize LAC genes were categorized into Group I, Group III, Group IV, and Group V, consistent with the phylogenetic analysis results of the LAC gene families in *Arabidopsis* [17], rice [23], soybean [12], and sorghum [24]. Within the genetic makeup of the LAC gene, the different categories included the three typical sections present in laccases: Cu-oxidase-3, Cu-oxidase-1, and Cu-oxidase-2. Each subgroup’s gene structure was consistent with the motif, so the gene family members remained intact. In spite of the more conserved copper ion binding domain in maize laccases, the genetic distance between these laccase members suggested that the have diverse functions.

The LAC genes in different subfamilies have different functions. In Group I, *ZmLAC1*, *ZmLAC7*, *ZmLAC8*, *ZmLAC9*, *ZmLAC18*, and *ZmLAC20* are closely related to *AtLAC17.* In *Arabidopsis*, laccases such as *AtLAC4*, *AtLAC11*, and *AtLAC17* play a crucial role in the polymerization of lignin, contributing significantly to lignin biosynthesis [89]. According to our phylogenetic tree, the first group of laccase members in maize has similar functions to the second group. This means that *ZmLAC1*, *ZmLAC7*, *ZmLAC8*, *ZmLAC9*, *ZmLAC18*, and *ZmLAC20* increase the possibility of catalyzing lignin biosynthesis. Since *AtLAC4* and *AtLAC17* are located in the secondary cell wall, it is likely that *ZmLAC18* is involved in the mitochondrial redox cycle or similar processes due to its expected presence in the mitochondria [89,90]. *AtLAC3*, *AtLAC5*, *AtLAC12*, and *AtLAC13* respond to the abscisic acid (ABA) signal [17,91]. *ZmLAC2*, *ZmLAC10*, *ZmLAC16*, and *ZmLAC21* contain ABA elements in their promoter. Under conditions of drought, low temperature, salt, or osmotic stress, plants quickly accumulate ABA, leading to increased resistance.

Interestingly, Group IV contained two maize LAC genes (*ZmLAC15* and *ZmLAC17*). During biotic stress, *AtLAC6* is inhibited; however, *AtLAC15* is involved in the oxidative polymerization of flavonoids in the seed coat of *Arabidopsis* [32]. These enzymes have similar low redox potentials and have been previously recognized for their involvement in the polymerization of phenolic compounds, which may be associated with environmental factors like drought, low temperature, and salt stress. *ZmLAC3, ZmLAC4, ZmLAC5*, *ZmLAC6*, *ZmLAC11*, *ZmLAC12*, *ZmLAC13*, *ZmLAC14*, *ZmLAC19*, and *ZmLAC22* were categorized in Group V alongside stress-triggered *AtLAC7*, *AtLAC8*, and *AtLAC9,* as well as *SbLAC4*, *SbLAC21*, and *SbLAC22*, potentially participating in the reaction to Cadmium stress [12,24]. The hypothesis of a high-redox-potential plant laccase revealed the response of maize laccase to environmental stress.

Moreover, the primary causes of plant gene family expansion are fragment and tandem repeats [78,79,80,81,82,83,84,85,86,87,88,89,90,91,92,93,94]. The examination of the recurring pattern of the maize LAC gene group uncovered two sets of adjacent duplicate genes and nine sets of fragment duplicate genes in maize, suggesting that the expansion of the maize LAC genes is primarily driven by fragment duplications. An interspecies covariance analysis of the LAC gene families from *Arabidopsis*, maize, sorghum, rice, and soybean revealed that 17 LAC genes in maize were covariant with *Arabidopsis*, rice, soybean, and sorghum, with a total of 96 interspecific covariants; the maize LAC genes were covariant with 8, 37, 28, and 23 covariants of *Arabidopsis*, soybean, rice, and sorghum, respectively, of which *ZmLAC2*, *ZmLAC10* both contained the most covariates, at 14, and *ZmLAC21* contained 12 covariates, the second most. *ZmLAC3*, *ZmLAC5*, *ZmLAC8*, *ZmLAC9*, *ZmLAC12*, and *ZmLAC14* had no covariate relationships with these species. These results suggest that the mean Ka, Ks, and Ka/Ks ratios for adjacent duplications in maize LAC genes were higher compared to those of the segmental duplicated genes, except for maize genes matched with *Arabidopsis* and rice homologous genes, whose Ks values exceeded 2. Conversely, the maize gene pairs showed Ka, Ks, and Ka/Ks values below 1 when compared to homologous gene pairs in *Arabidopsis*, soybeans, rice, and sorghum. The duplication events of the LAC gene tandem and fragment repeat genes in maize are believed to have taken place around 39.83 and 33.37 million years ago, with maize diverging from *Arabidopsis*, soybean, rice, and sorghum approximately 212.55, 228.93, 39.32, and 30.367 million years ago, respectively. The reason for this is that maize, rice, and sorghum are monocotyledonous gramineous plants with a close genetic connection, while *Arabidopsis* and soybean are dicotyledonous plants with a distant genetic relationship.

Due to the latest developments in high-throughput sequencing technology, the cost of transcriptome sequencing has been gradually decreasing [95]. However, although high-throughput sequencing technology has been gaining popularity, it is still important to verify its results. Several researchers have sequenced maize transcriptomes, resulting in a large dataset. Utilizing big data from transcriptome sequencing effectively can lead to cost savings and further exploration of the data. Moreover, the analysis of maize gene families’ expression profiles under various conditions can be conducted using transcriptome sequencing data. Publicly available maize transcriptome sequencing data and maize B73_V5 genomic data were used in this study to analyze the tissue-specific expression patterns of 22 maize LAC gene families and their reactions to various biotic and abiotic stressors. The maize LAC gene family exhibited high levels of expression prior to the V9 Thirteenth Leaf stage, but showed minimal expression afterwards. *ZmLAC7*, *ZmLAC8*, *ZmLAC9*, *ZmLAC10*, *ZmLAC18*, *ZmLAC19*, and *ZmLAC20* exhibited peak expression levels in the V9 Eleventh Leaf, V9 Thirteenth Leaf, and V9 Immature Leaves phases. Maize was in its nodulation stage at this time, with the maize stem nodes elongating upwards and requiring significant laccase catalysis to facilitate xylem synthesis. An analysis of the maize LAC gene family’s expression patterns in reaction to abiotic and biotic stress revealed that four maize LAC genes (*ZmLAC1*, *ZmLAC11*, *ZmLAC12*, and *ZmLAC16*) showed no expression in either scenario. The results suggest that the LAC gene family is involved in regulating plant growth and development, in line with the findings from tissue-specific expression analysis.

In this investigation, all maize LAC gene families, except *ZmLAC1*, *ZmLAC11, ZmLAC12*, *ZmLAC13*, *ZmLAC14*, *ZmLAC16*, and *ZmLAC21,* exhibited significant responses to at least one form of stress. Specifically, *ZmLAC2*, *ZmLAC3*, *ZmLAC4*, and *ZmLAC18* demonstrated significant differential expression exclusively under abiotic stress conditions. In contrast, *ZmLAC6*, *ZmLAC8*, and *ZmLAC20* exhibited significant differential expression only under biotic stress. Additionally, *ZmLAC5*, *ZmLAC7*, *ZmLAC9*, *ZmLAC10*, *ZmLAC15*, *ZmLAC17*, *ZmLAC19*, and *ZmLAC22* showed significant differential expression in response to both abiotic and biotic stress factors.

Many plant species have also been reported to respond to unfavorable environmental conditions by expressing LAC genes. For example, the maize LAC genes *ZmLAC5*, *ZmLAC10*, *ZmLAC15*, *ZmLAC17*, and *ZmLAC19* are involved in the salt stress response. Following exposure to abiotic stressors like salt, drought, or heavy metals, rice *OsLAC10* facilitated the conversion of monolignol into lignin, leading to increased copper tolerance and reduced copper buildup in plant roots through the upregulation of *OsLAC10* [23]. The genes *ZmLAC4* and *ZmLAC10* are involved in the response to drought stress in maize, while *AtLAC2* in *Arabidopsis* is involved in the response to dry conditions. Research on LACs primarily centers around studying the function of the enzymes responsible for encoding genes related to lignin production and their response to different stress factors [96]. Additionally, there is proof indicating that LAC genes contribute to the ability of plants to resist stress. LAC genes are expressed in a variety of plants and they function similarly in response to different kinds of stresses. The response of the LAC genes in different plants to biotic stresses is similar. For example, maize LAC genes, including *ZmLAC5*, *ZmLAC9*, and *ZmLAC10,* respond to diseases. In a study on cotton laccase genes, the *GhLAC15* gene played a role in lignin production and was closely linked to verticillium wilt resistance [40]. The upregulation of LAC genes contributes to bolstering plant defense mechanisms against pathogens and pests, but this outcome is not solely reliant on the augmentation of lignin levels. At the same time, LAC influences the synthesis of phenolic substances in plants and alters the metabolic pathway of phenylpropanoids, resulting in the accumulation of jasmonic acid and additional compounds while reducing the effects of toxic substances [38,97,98].

The maize LAC gene group showed varying expression patterns when exposed to different abiotic and biotic stress factors. Specifically, four genes (*ZmLAC2*, *ZmLAC3*, *ZmLAC4*, and *ZmLAC18*) were uniquely expressed under abiotic stress, while three genes (*ZmLAC6*, *ZmLAC8*, and *ZmLAC20*) were specifically responsive to biotic stress. Furthermore, seven genes (*ZmLAC5*, *ZmLAC7*, *ZmLAC9*, *ZmLAC10*, *ZmLAC17*, *ZmLAC19*, and *ZmLAC22*) were differentially expressed under both stress conditions, albeit with distinct expression profiles. *ZmLAC15* exhibited a notable upregulation in expression levels in response to both stress conditions, suggesting its crucial involvement in stress response and positioning it as a potential gene of interest for future research. The seven maize LAC genes that were left did not show differential expressions in response to either stress condition. The analysis of maize LAC genes under different stress conditions provides valuable insights for future molecular biology research.

The results indicate that LAC genes have a consistent role in different plant species when dealing with both biotic and abiotic stress. The *ZmLAC5* gene showed unique expression profiles in response to factors like waterlogging, temperature changes, gray spot disease, and smut infection. *ZmLAC17* exhibited different levels of expression when exposed to salt stress, waterlogging, temperature changes, and smut infection, whereas *ZmLAC10* showed differential expressions in reaction to salt stress, flooding, smut, and infestation by beet armyworm. *ZmLAC5*, *ZmLAC10*, and *ZmLAC17* showed notable changes in expression levels in response to all four biotic and abiotic stress factors, indicating their important functions in stress adaptation and that they deserve more research as potential key genes. This research has uncovered new understanding of how LAC genes may help alleviate both biotic and abiotic stress in maize, allowing for a better understanding of the molecular processes involved in maize’s adaptation to environmental challenges.

## 5. Conclusions

This study identified 22 LAC genes in maize using its genome information. After examining their physicochemical characteristics, chromosome positioning, gene organization, evolutionary relationships, and collinearity, it was determined that the 22 maize LAC gene families were spread across four subfamilies on nine chromosomes. Every subgroup exhibited an identical gene structure and motif, with all gene family members being intact and preserved. The transcriptome sequencing data were reanalyzed using bioinformatics methods and the published maize B73_V5 transcriptome sequencing dataset. An investigation was carried out on how the maize LAC gene family is expressed in various tissues and how it reacts to stress. Its expression patterns were different and coordinated the growth and development of maize. *ZmLAC5*, *ZmLAC10*, and *ZmLAC17* showed differential expressions when exposed to different abiotic and biotic stress factors. This study offers valuable insights for analyzing the function and evolution of maize LAC genes, presenting potential candidate genes that could enhance maize stress resistance.

## Figures and Tables

**Figure 1 genes-15-00749-f001:**
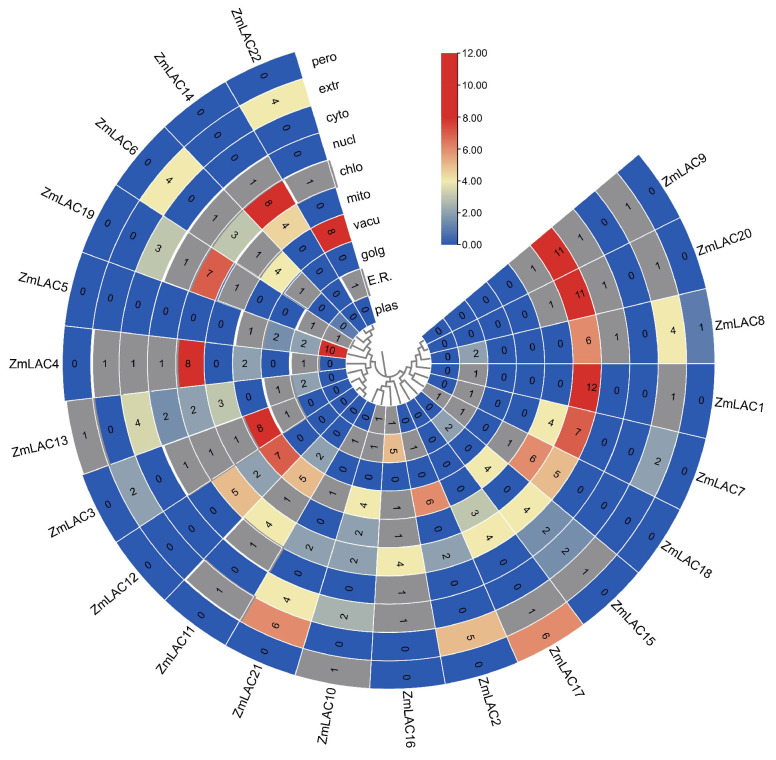
Heat map showing where *ZmLAC* genes are located in plants, including the plasmids, endoplasmic reticula, Golgi apparatus, vacuoles, mitochondria, chloroplasts, nuclei, cytosols, extracellular matrix, and peroxisomes of the plant cell. The absence of the relevant gene in the specified area is shown by a blue color, while the minimal functional presence of the gene is indicated by grey, and the maximum functional significance of the gene is represented by red. Note: E.R.: Endoplasmic Reticulum; golg: Golgi apparatus; Vacu: vacuole; mito: mitochondria; chlo: chloroplast; nucl: nucleus; cyto: cytosol; extr: extracellular matrix; pero: peroxisome.

**Figure 2 genes-15-00749-f002:**
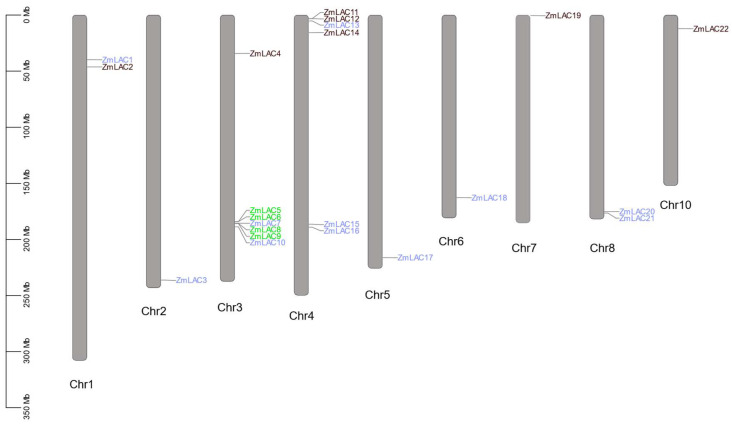
Genes belonging to the LAC family are spread out over various maize chromosomes. Note: green genes are tandem duplications, blue genes are segmental duplications.

**Figure 3 genes-15-00749-f003:**
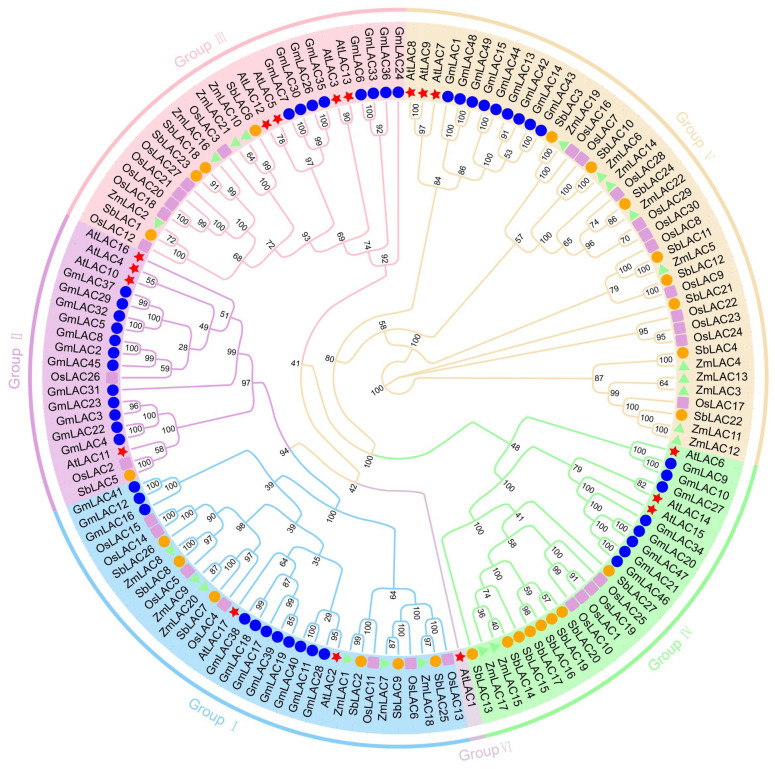
Analyzing phylogenetic relationships of LAC proteins with maize, *Arabidopsis*, soybean, sorghum, and rice.

**Figure 4 genes-15-00749-f004:**
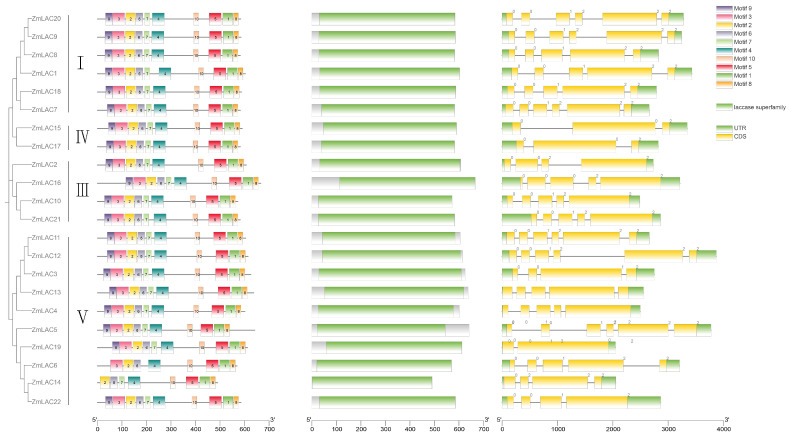
Schematic diagram of conserved protein motif, domain, and exon–intron structure of LAC gene in maize.

**Figure 5 genes-15-00749-f005:**
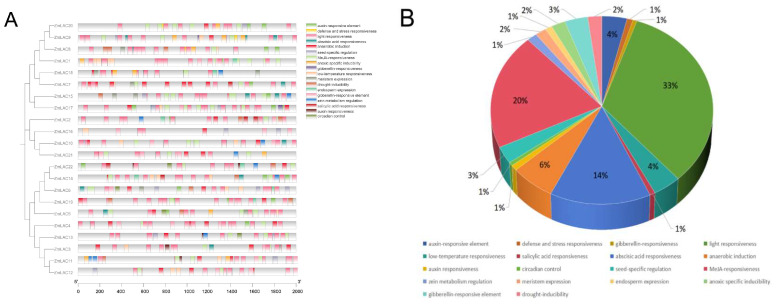
Investigating the cis-regulation factors that affect the promoters of maize LAC family genes. (**A**) Different cis-regulatory elements are spread out. (**B**) The cis-acting elements that make up the promoters of maize LAC genes vary in their relative proportions.

**Figure 6 genes-15-00749-f006:**
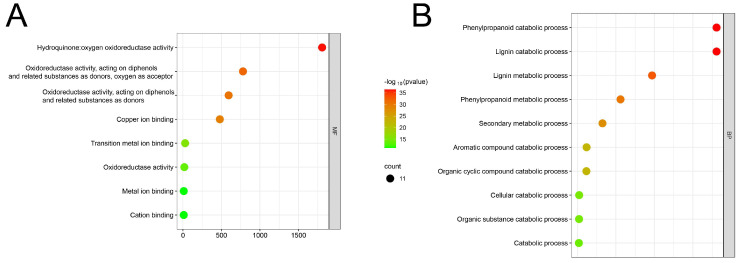
A fold enrichment chart representing the overlap of functions between the *ZmLAC* genes. (**A**) GO molecular function; (**B**) GO biological function. Note: Red dot plots indicate the most genes involved in that process, while small green plots indicate the least genes involved.

**Figure 7 genes-15-00749-f007:**
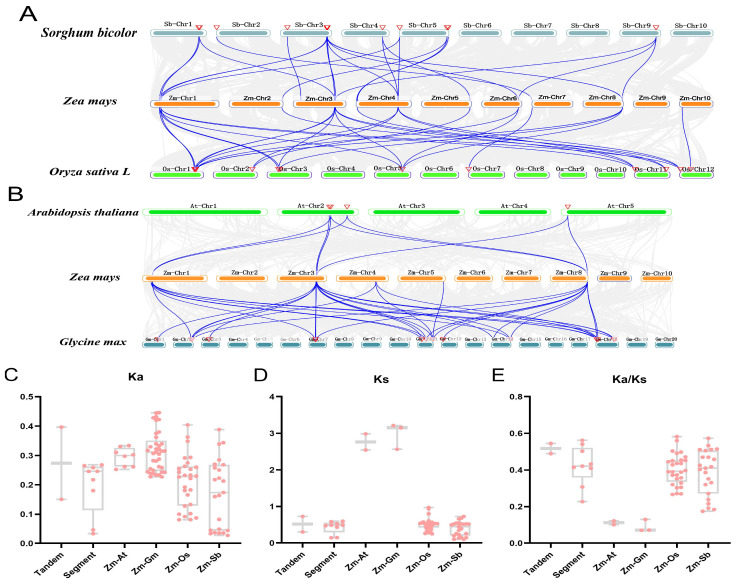
Gene duplication and homolinearity of the LAC gene in maize. (**A**): The LAC gene of maize is related to dicotyledonous plants (*Arabidopsis*, soybean), and their gene pairs are connected with the blue lines; (**B**): Monocotyledonous plants (rice, sorghum) with homogeneic relationship, and their gene pairs connected with the blue line. The horizontal axes in (**C**–**E**) represent tandem repeats, segment repeats, and repeats between corn and *Arabidopsis* (Zm-At), soybean (Zm-Gm), rice (Zm-Os), and sorghum (Zm-Sb), respectively.

**Figure 8 genes-15-00749-f008:**
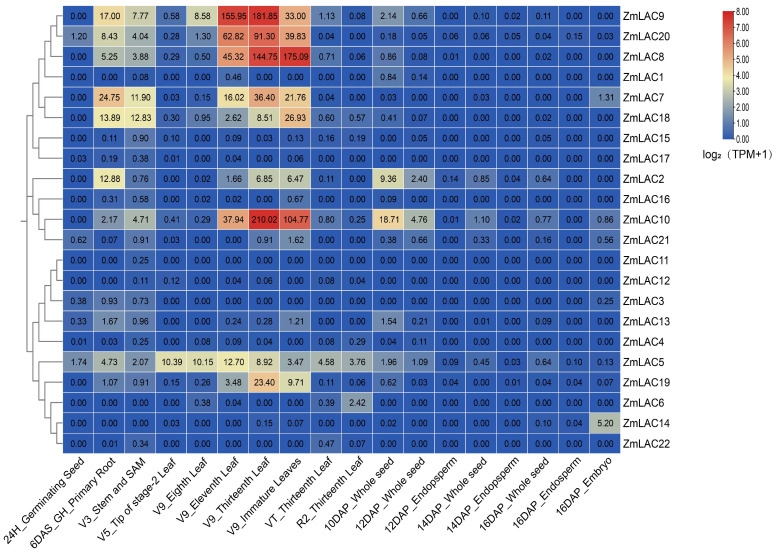
Maize LAC gene family expression heat map in different tissues. Note: The data in the boxes indicate original TPM values.

**Figure 9 genes-15-00749-f009:**
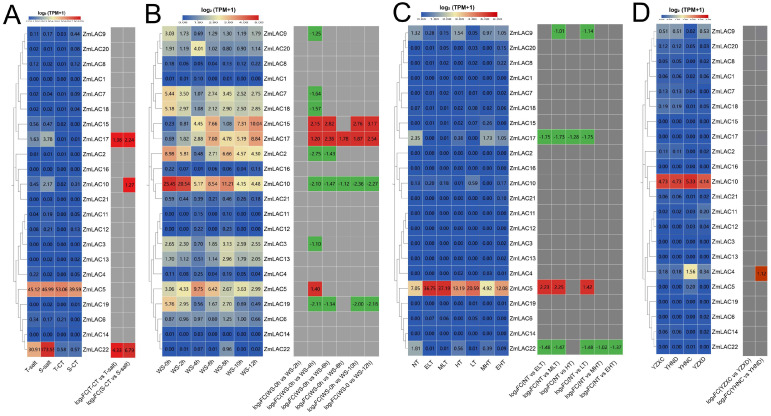
Heat map showing the expression of the LAC gene family in maize when exposed to environmental stress. (**A**) Salt stress expression pattern of LAC genes in maize. T—salt: inbred salt-tolerant maize line L87 (220 mM NaCl) treated with salt; S-salt: inbred line L29 treated with salt (220 mM NaCl) for salt sensitivity; T—CT: inbred salt-tolerant maize line (L87) (0 mM NaCl) treated as control; S—CT: inbred salt-sensitive maize line (L29) (0 mM NaCl) treated as control. (**B**) The maize LAC gene family’s expression pattern under flooding stress. WS—0 h: 0 h; WS—2 h: 2 h; WS—4 h: 4 h; WS—6 h: 6 h; WS—8 h: 8 h; WS—10 h: 10 h; WS—12 h: 12 h. (**C**) Affect of high- and low-temperature stress on maize LAC gene expression pattern. EHT: very high temperature (48 °C); MHT: moderately high temperature (42 °C); HT: elevated temperature (37 °C); NT: standard temperature (25 °C); LT: reduced temperature (16 °C); ELT: very low temperature (4 °C); MLT: moderate and reduced temperature (10 °C). (**D**) The LAC gene family’s expression pattern in maize was analyzed under drought stress conditions using drought-intolerant hybrid (ZX978) samples treated with water, drought-tolerant hybrid (ND476) samples treated with drought, drought-tolerant hybrid (ND476) samples treated with water, and drought-resistant hybrid (ZX978) samples treated with drought. Note: there are two boxes in each chart—the left box represents the original TPM and the right box represents the log 2 (multiples change) indicated in red for up-regulation and green for down-regulation, |FC| ≥ 1.

**Figure 10 genes-15-00749-f010:**
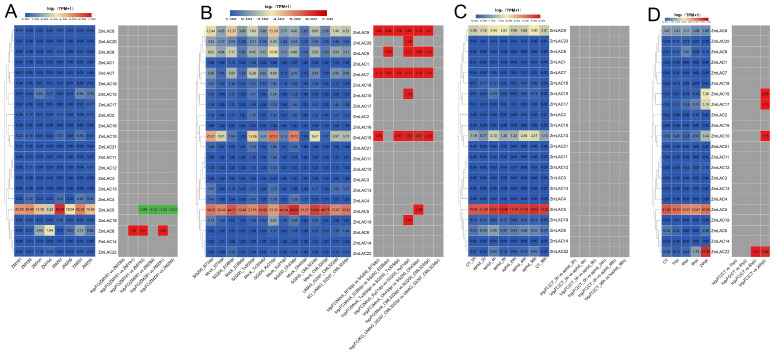
Heat map showing the expression of the LAC gene family in maize during biological stresses. (**A**) An investigation was conducted on the LAC gene family expression in maize plants affected by gray spot disease, with two varieties, ZMY (‘Yayu 889’) and ZMZ (‘Zheng Hong 532’), being exposed to the disease and infected at 81, 89, 91, and 93 days post planting. (**B**) The LAC gene family’s expression pattern in maize under smut stress. SG 200 and UMAG_02297 are strains of the active vegetative fungus maize melitrogen, which leads to maize smut. KO_UMAG_02297 is a mutant strain of UMAG_02297. The simulation control treatment involves uninfected plants. The term 3dpi refers to 3 days after infection. The six maize lines are B73, CML 322, EGB, Ky 21, Oh 43, and Tx 303. (**C**) The maize LAC gene family’s expression pattern was examined under aphid stress, with CT-0h and CT-96h serving as control treatments or uninfected plants at 0 h and 96 h. Following aphid infestation, time intervals of 2 h, 4 h, 8 h, 24 h, 48 h, and 96 h were marked as Aphid—2 h, Aphid—4 h, Aphid—8 h, Aphid—24 h, Aphid—48 h, and Aphid—96 h, correspondingly. (**D**) The maize LAC gene family’s expression patterns were observed under the beet moth stress, with CT, the control treatment, representing uninfected plants. The time points of 1 hpi, 4 hpi, 6 hpi, and 24 hpi corresponded to 1, 4, 6, and 24 h post-infection, respectively. Note: there are two boxes in each chart—the left box represents the original TPM and the right box represents the log 2 (multiples change), indicated in red for up-regulation and green for down-regulation, |FC| ≥ 1.

**Figure 11 genes-15-00749-f011:**
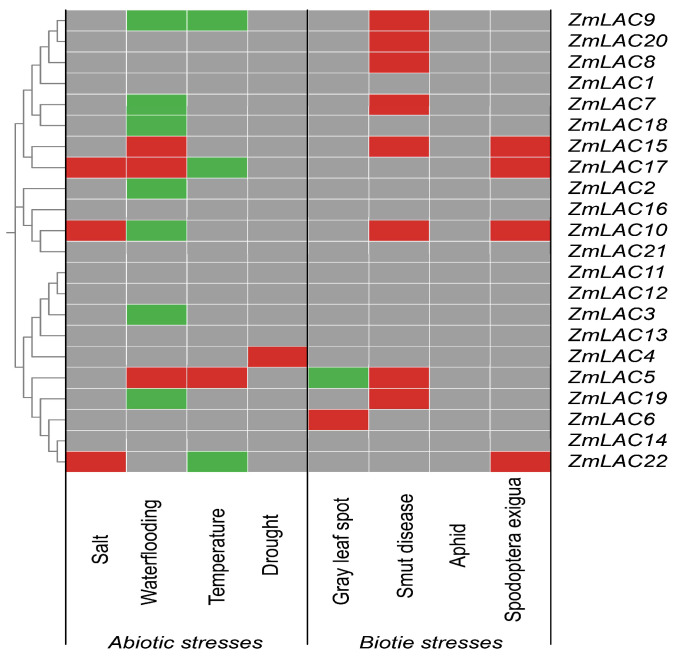
The maize LAC gene family’s expression pattern in reaction to abiotic and biotic pressures is displayed as a heat map. A gray hue signifies no alteration in expression, a crimson hue signifies an escalation in expression, a verdant hue signifies a reduction in expression, and a cerulean hue signifies both an increase and decrease in expression.

## Data Availability

Data are contained within the article and Appendix A.

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
