# Peer review of "The Analysis, Description, and Examination of the Maize LAC Gene Family’s Reaction to Abiotic and Biotic Stress"

_genes, 2024, doi:10.3390/genes15060749_

Round 1

Reviewer 1 Report

Comments and Suggestions for Authors
  1. Please include the citation for web servers such as Pfam, HMME, InterPro, etc., and the release version of software/packages used (BLASTP, MEME, TBtools, SRA toolkit, etc.).
  2. Considering that the LAC gene family is also found in rice, sorghum, and soybean, why were only Arabidopsis LAC genes used for BLAST searches to identify LAC genes in maize? Would incorporating LAC genes from these additional species (rice and sorghum being monocots) enhance the identification and functional annotation of maize LAC genes?
  3. The results of the manuscript are primarily based on computational approaches. The significance of identified molecular signatures and biomarkers remains uncertain without experimental validation. The authors should consider including a section on the limitations of the study, acknowledging 
    1. Lack of functional validations and potential consequences.
    2. Limitations of a sequence homology-based approach: The approach might miss novel LAC genes with low sequence similarity or include non-LAC genes with high similarity in certain domains.
  1. In Supplementary Table 1, 
    1. BLAST hit information is not available. Including this information helps to understand the sequence conservation and potential functional similarities. For each laccase candidate gene, indicate the best blast hits in Arabidopsis for the laccase candidates, specifically the top hits in Arabidopsis laccase genes. Also, include the gene ID, E-value, and sequence identity percentage for each best BLAST hit.
    2. Include chromosome coordinates of the laccase candidate genes.
  1. Please include the information about copper oxidase domains (cu-oxidase-1/2/3) detected in each candidate, including the start and end positions within the protein sequence, in the supplementary file.
  2. In section 3.3, please include supplementary information regarding identified orthologous and paralogous relationships. 
  3. In supplementary file 4, include the E-value of any motifs discovered by MEME.
  4. “Three LAC genes (ZmLAC10, ZmLAC17, and ZmLAC22) were significantly upregulated under salt stress in salt-sensitive and salt-tolerant maize compared to their controls.” In 3.8 and 3.9, explain how the statistical significance was observed/computed for the “significantly upregulated” and “significantly downregulated” cases mentioned. 
  5. Section 3.10: Please provide the details of the methods for performing differential analysis. 

Reviewer 2 Report

Comments and Suggestions for Authors

Dear appreciated Authors,

The manuscript:” Identification, Characterization, and Expression Profiling of the 2 Maize LAC Gene Family in Response to Abiotic and Biotic 3 Stressors” ) genes-3028259-peer-review-v1.pdf) is good, the summary is well written, the material and methods are well-planned, the Results and discussion in detail explained and the Conclusion is appropriate. The study provides valuable insights insight into the roles of LAC genes in the abiotic and biotic stress responses and provides a theoretical foundation for understanding the molecular mechanisms underlying maize adaptation to unfavorable conditions. It was confirmed different expressions of ZmLAC5, ZmLAC10 and ZmLAC17 in response to various abiotic and biotic stressors.

Overall, this manuscript is well-written. There are some points where more accurate, or more detailed information would yield benefit. I have only a few comments below.

 1.      In Abstract, Lines 13: please “on” (instead on) - (However, a comprehensive investigation of maize lacase has 13 not yet been documented).

2.      In Section 1 Introduction, Line33: Put the word "has" instead of "have"

3.      Line 37: T1 - if it's the first time a mention appears in the text, it is useful to write the full name and abbreviation in brackets.

4.      Line 52: New sentence: However, recent studies suggest that knowledge ,regarding LAC in higher plants is still limited (put references)

5.      Line 64: Input the other, alternative way to improve maize productivity and response, ex. Despite the clear advantages of newly developed varieties, the productive ability of a maize genotype has been still limited.  A variety of factors, including climate, pests, soil properties, solar radiation, field management practices, seed quality, and genetic potential, can significantly impact a maize genotype's productivity and yield potential (Barošević et al., 2022).

Reference example: Barošević, T.; Bagi, F.; Savić, Z.; Ljubičić, N.; Ivanović, I. Assessment of Maize Hybrids Resistance to Aspergillus Ear Rot and Aflatoxin Production in Environmental Conditions in Serbia. Toxins 2022, 14, 887. https://doi.org/10.3390/toxins14120887

Line 246: Please set up appropriate figure dimensions – Figure 2.

Line 556: Due to the latest developments in high-throughput sequencing technology, the cost of transcriptome sequencing has been gradually decreasing [81]. However, beside high-throughput sequencing technology has gaining popularity but it is  still important to verify results.

In general, the manuscript is well written.
